# Push, Pop, Parallelize: Stack-Augmented Linear Attention via the Delta Rule

Anh T Nguyen [1]   Saleh Momeni [1]   Ashutosh Chaubey [2]   Changnan Xiao [3]   Bing Liu [1]

## Abstract

Linear attention architectures based on the Delta rule, such as DeltaNet and RWKV-7, combine Transformer-level performance with RNN-like efficiency and provably solve regular language tasks. However, their fixed-size states struggle to capture the recursive, hierarchical structures intrinsic to natural languages. To bridge this gap, we introduce DeltaStack, which augments DeltaNet's associative memory with a lightweight, differentiable stack. Unlike prior approaches that rely on sequential recurrence, DeltaStack formulates stack operations as linear delta-rule updates, enabling a hardware-aware implementation fully parallelizable over sequence length. Theoretically, we prove DeltaStack extends DeltaNet's expressivity to model both regular and hierarchical languages. Empirically, DeltaStack outperforms DeltaNet and Stack-Attention on formal language benchmarks and consistently surpasses DeltaNet baselines in language modeling perplexity and zero-shot performance across scales up to 760M parameters. Our code is publicly available at https://github.com/teeann/DeltaStack.

## 1. Introduction

The Transformer architecture (Vaswani et al., 2017) has firmly established itself as the backbone of modern large language models (LLMs). Its success is largely attributed to the massively parallelizable training of the self-attention mechanism, which allows for scaling to unprecedented dataset sizes (Brown et al., 2020; Team et al., 2023; Touvron et al., 2023). However, despite their dominance, Transformers suffer from two fundamental limitations. First, the quadratic complexity of self-attention and the linear growth of the Key-Value (KV) cache result in prohibitive memory and I/O costs during inference on long sequences (Dao et al., 2022). Second, from a computational complexity perspective, standard Transformers are limited to the circuit complexity class $\mathsf{TC}^0$, rendering them theoretically incapable of solving state-tracking tasks or modeling processes that require sequential reasoning depth greater than the network's depth (Merrill et al., 2022; 2024).

**The Hierarchy Gap.** While recent linear attention models (Grazzi et al., 2025; Siems et al., 2025; Yang et al., 2024b) excel at regular languages and state tracking, they fundamentally struggle to capture the *recursive, hierarchical structures* intrinsic to human language and reasoning. Such limitation to model Deterministic Context-Free Languages (DCFLs) (Merrill, 2019; Bernardy, 2018; Hewitt et al., 2020) is even more acute in formal reasoning domains such as mathematics and code generation, where understanding nested scopes (e.g., matching parentheses, loops inside loops) is a prerequisite for correctness. A fixed-size state cannot theoretically resolve unboundedly deep recursion (Merrill, 2019; Bernardy, 2018; Hewitt et al., 2020; Malach et al., 2026), effectively capping the reasoning ceiling of current linear architectures.

**The Challenge of Scalable Stacks.** Bridging this gap requires equipping the model with unboundedly growing memory, typically in the form of a stack. However, integrating a stack into modern neural architectures presents a difficult dilemma between *expressivity* and *efficiency*. Naive implementations of stack-augmented RNNs (Joulin & Mikolov, 2015; DuSell & Chiang, 2024; Li et al., 2024a) rely on sequential push/pop operations that destroy the parallel training capabilities essential for modern scale. Conversely, recent attempts to approximate stacks within attention mechanisms (DuSell & Chiang, 2024; Zhang et al., 2026) often resort to ad-hoc heuristics that fail to provide rigorous guarantees for DCFL recognition. The open challenge, therefore, is to design a memory mechanism that is strictly expressive enough to capture hierarchy yet able to preserve hardware-aware parallel training.

**Our Approach.** In this paper, we introduce **DeltaStack**, a novel architecture that resolves this tension by augmenting the associative memory of DeltaNet (Schlag et al., 2021; Yang et al., 2024b) with a lightweight, differentiable stack. Crucially, rather than treating the stack as a separate sequen-

[1]University of Illinois Chicago [2]University of Southern California [3]ChangnXX.github.io. Correspondence to: Bing Liu <liub@uic.edu>.

*Proceedings of the 43$^{rd}$ International Conference on Machine Learning*, Seoul, South Korea. PMLR 306, 2026. Copyright 2026 by the author(s).

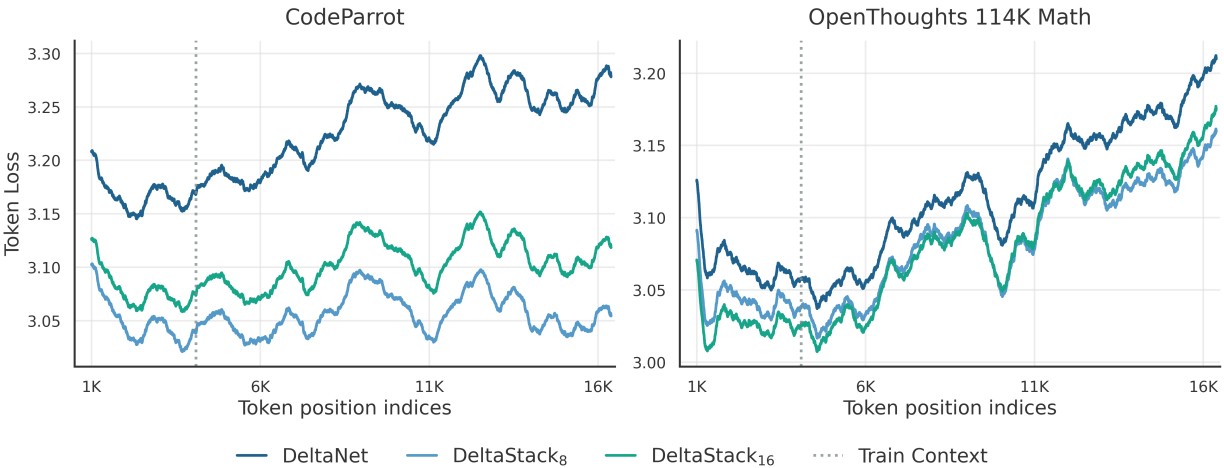

*Figure 1.* **Length Extrapolation on Reasoning Benchmarks.** Per-token loss for 340M models trained on 15B tokens with 4096 context length. DeltaStack robustly extrapolates beyond its training length, showing substantial gains in math and coding tasks.

tial module, we formulate stack operations-push, pop, and read-as *linear delta-rule updates* to a specialized memory state matrix. This formulation allows us to leverage the highly efficient, chunkwise-parallel algorithms developed for linear transformers, enabling our stack to be trained in parallel across the sequence length.

Our contributions are as follows:

- **Architecture:** We propose DeltaStack, a modular extension to linear attention that integrates a differentiable stack via a novel moving-pointer mechanism. By casting stack dynamics as linear state-space updates, our method maintains the hardware-aware parallel training efficiency of DeltaNet ($O(LCd)$ complexity) while enabling unbounded state tracking.

- **Theory:** We provide a theoretical characterization of the model's expressivity, proving that DeltaStack extends the capabilities of standard DeltaNet to recognize both Regular Languages and Deterministic Context-Free Languages (DCFLs).

- **Formal Languages Performance:** On comprehensive formal language benchmarks, DeltaStack outperforms the second-best linear attention baselines by an average of 14% across 8 tasks.

- **Language Modeling & Extrapolation:** We validate our approach on 340M-parameter language models trained on 15B tokens. DeltaStack achieves perplexity competitive with strong DeltaNet baselines while demonstrating superior length generalization on Code-Parrot and OpenThoughts-114k-Math benchmarks.

## 2. Related work

**Expressivity Limits and Stack Augmentation.** Despite the dominance of the Transformer architecture, theoretical analyses demonstrate its inherent limitations in modeling the recursive and hierarchical structures characteristic of the Chomsky hierarchy, particularly Deterministic Context-Free Languages (DCFLs) (Hahn, 2020; Deletang et al., 2023). To bridge this gap, prior research has explored augmenting neural networks with external stack memory, evolving from early Stack-RNNs (Joulin & Mikolov, 2015; Grefenstette et al., 2015) to recent Stack-Transformers (DuSell & Chiang, 2024; Li et al., 2024a; Zhang et al., 2026). However, these existing approaches often face a dilemma: they either rely on sequential stack operations that break training parallelism (DuSell & Chiang, 2024; Li et al., 2024a), or introduce ad-hoc layerwise stacks that fail to guarantee DCFLs expressiveness (Zhang et al., 2026). DeltaStack distinguishes itself by integrating the stack mechanism directly into the *linear* attention framework. By formulating stack push/pop operations as parallelizable Delta-rule updates, DeltaStack captures the expressivity benefits of explicit stack memory without sacrificing the hardware efficiency essential for modern large-scale training.

**Length Generalization.** Despite the dominance of Transformer-based LLMs, they often exhibit catastrophic failure when processing sequences longer than those encountered during training (Zhou et al., 2024). This challenge has spurred extensive research, ranging from theoretical analyses of Transformer expressivity and learnability (Xiao & Liu, 2025; Abbe et al., 2024; Awasthi et al.; Huang et al., 2025; Izzo et al., 2026) to empirical improvements such as novel positional embeddings (Cho et al., 2025; Li et al., 2024b), specialized training curricula (Lee et al., 2025), modified

input formats (Lee et al., 2024), and tool-based agentic settings (Malach et al., 2026). A complementary line of work investigates the specific classes of formal languages that neural architectures can systematically recognize or generate (Deletang et al., 2023; Butoi et al., 2025). In Section 5.1, we demonstrate that DeltaStack achieves robust length generalization on both Regular and Deterministic Context-Free (DCF) tasks, effectively maintaining performance on sequences far exceeding the training context.

**Linear Attention.** Standard linear attention mechanisms (Katharopoulos et al., 2020) reduce the quadratic complexity of Transformers to linear time by removing the softmax operation and decoupling the query and key interactions. This allows the model to compress the sequence of past tokens into a fixed-size, flat memory state that updates recurrently, enabling $O(1)$ inference complexity similar to RNNs. To overcome the capacity limitations of additive updates of standard linear attention (Schlag et al., 2021; Yang et al., 2024a), recent architectures incorporate the Delta rule (Widrow & Hoff, 1988) to enable selective overwriting of the memory state. Furthermore, the state update can also be formulated as a step of online gradient descent (test-time training) to minimize a linear regression objective (Schlag et al., 2021; Sun et al., 2025; Behrouz et al., 2025; Hu et al., 2026; Zhang et al., 2024; Yang et al., 2025; Siems et al., 2025). Models such as DeltaNet (Yang et al., 2024b) and RWKV-7 (Peng et al., 2025) leverage this data-dependent decay to achieve stronger associative recall while maintaining parallelizability.

## 3. Background

### 3.1. Stack-Augmented Recurrent Neural Networks

The Stack-Augmented Recurrent Neural Network (Joulin & Mikolov, 2015) (Stack RNN) extends standard recurrent networks with a differentiable stack memory. The model consists of a controller (implemented as a standard RNN) and an external stack $j_t$. At each time step $t$, the controller computes a hidden state $h_t$ and decides on an action to manipulate the stack. The system is made fully differentiable by defining stack operations as continuous value updates.

### 3.2. DeltaNet

DeltaNet (Yang et al., 2024b) advances standard linear attention architectures by introducing a *delta rule* update mechanism. Unlike the purely additive updates of standard linear transformers (Katharopoulos et al., 2020), the delta rule enables the model to selectively remove or rewrite information in its memory state. Given a sequence of input vectors $\mathbf{x}_1, \ldots, \mathbf{x}_L \in \mathbb{R}^d$, the model first computes queries, keys, and values via linear projections:

$$\mathbf{q}_t, \mathbf{k}_t, \mathbf{v}_t = \mathbf{W}_Q \mathbf{x}_t, \mathbf{W}_K \mathbf{x}_t, \mathbf{W}_V \mathbf{x}_t,$$

where $\mathbf{W}_{\{Q,K,V\}} \in \mathbb{R}^{d \times d}$. The core of DeltaNet is the evolution of its memory matrix $\mathbf{S}_t \in \mathbb{R}^{d \times d}$. The state update and output computation are defined as:

$$\mathbf{S}_t = \mathbf{S}_{t-1} - \beta_t (\mathbf{S}_{t-1} \mathbf{k}_t - \mathbf{v}_t) \mathbf{k}_t^\top,$$
$$\mathbf{o}_t = \mathbf{S}_t \mathbf{q}_t$$

where $\beta_t \in \mathbb{R}$ is a data-dependent step size with values in range $[0, 2]$ (Grazzi et al., 2025). This update rule can be interpreted as performing a single step of Stochastic Gradient Descent (SGD) to minimize an online linear regression objective (Sun et al., 2025; Behrouz et al., 2025):

$$\mathcal{L}_t(\mathbf{S}) = \frac{1}{2} \| \mathbf{S} \mathbf{k}_t - \mathbf{v}_t \|^2.$$

By rearranging the terms, the recurrence reveals a linear sequential structure:

$$\mathbf{S}_t = \mathbf{S}_{t-1}(\mathbf{I} - \beta_t \mathbf{k}_t \mathbf{k}_t^\top) + \beta_t \mathbf{v}_t \mathbf{k}_t^\top. \tag{1}$$

The multiplicative term $\mathbf{P}_t = (\mathbf{I} - \beta_t \mathbf{k}_t \mathbf{k}_t^\top)$ acts as a generalized Householder reflection applied to the previous state. This structure offers two significant advantages:

**1. Efficient Parallel Training.** By leveraging the compact WY representation of Householder matrices (Bischof & Van Loan, 1987), the cumulative product of these transitions over a chunk of size $C$ can be computed efficiently. This enables a chunkwise-parallel training algorithm with a complexity of $O(LCd + Ld^2)$, preserving the parallelizability of Transformers while maintaining the inference efficiency of RNNs (Yang et al., 2024b).

**2. Theoretical Expressivity.** Recent theoretical work demonstrates that linear recurrent models parameterized with Diagonal-Plus-Low-Rank (DPLR) transitions such as DeltaNet, are capable of recognizing any regular language (Siems et al., 2025; Grazzi et al., 2025). This capability distinguishes DeltaNet from standard Transformers, as the ability to maintain and update a persistent state is critical for algorithmic reasoning tasks where tracking discrete state changes is required (Merrill et al., 2024).

## 4. Methodology

The main idea of our method is to introduce a stack-augmented parallelizable sequence processing architecture based on DeltaNet. Unlike several stack-augmented architectures that introduce a stack-like processing mechanism on top of transformer (Li et al., 2024a; DuSell & Chiang, 2024; Zhang et al., 2026), DeltaStack can theoretically handle both regular language and DCF tasks.

### 4.1. Stack memory with moving head pointer

We first introduce a novel differentiable stack memory, whose read and write operation can be parameterized as

matrix vector product. Given a stack $\mathbf{J}_t \in \mathbb{R}^{d \times s}$, where $s$ is the stack size and $d$ is the dimension of stack cells, we keep track of $p_t \in \mathbb{R}$, and use it to compute the corresponding vector pointer $\mathbf{p}_t \in \mathbb{R}^s$ that serves as a soft, differentiable index for accessing $\mathbf{J}_t$. At each time step $t$, $p_t$ is updated based on the model's action probabilities $\alpha_t \in \Delta^3$, where $\Delta^3$ denotes the probability simplex (i.e., the set of valid probability distributions) over the actions $\{\text{PUSH}, \text{POP}, \text{NO-OP}\}$. The position evolves as a scalar drift:

$$p_t = p_{t-1} + \alpha_t^{\text{push}} - \alpha_t^{\text{pop}} \tag{2}$$

$$\mathbf{p}_t = \phi(p_t) \tag{3}$$

The soft pointer vector $\mathbf{p}_t$ is then generated via a kernel $\phi$. Specifically, we employ a **Laplacian kernel** based on the $L_1$ distance of the stack indices, with a learnable scalar parameter $\gamma$. Let $i \in \{0, \dots, s-1\}$ denote the physical indices of the stack slots. The $i$-th component of the vector pointer $\mathbf{p}_t$ is computed via a softmax over negative $L_1$ distances:

$$\mathbf{p}_t[i] = \frac{\exp\left(-\gamma |i - p_t|\right)}{\sum_{j=0}^{s-1} \exp\left(-\gamma |j - p_t|\right)}$$

where $\gamma = \text{softplus}(\hat{\gamma})$ is a strictly positive, learnable scalar parameter representing the **sharpness** of the kernel. This parameter allows the model to dynamically adapt the granularity of its memory access: a high $\gamma$ approximates a discrete "hard" pointer for precise stack manipulation, while a lower $\gamma$ enables diffuse attention over a local neighborhood of stack slots.

With this formulation, the standard stack operations are cast as linear algebraic updates on the memory matrix $\mathbf{J}_t$:

**1. Read:** Retrieving the top of the stack is performed via a matrix-vector product between the current memory state and the pointer:

$$\mathbf{v}_t^{\text{read}} = \mathbf{J}_t \mathbf{p}_t \tag{4}$$

**2. Push:** To push a new value $\mathbf{v}_t^{\text{push}} \in \mathbb{R}^d$ onto the stack, we apply a Delta Rule update on $\frac{1}{2} \left\| \mathbf{J}\mathbf{p}_t - \mathbf{v}_t^{\text{push}} \right\|^2$, with step size $\alpha_t^{\text{push}}$.

$$\mathbf{J}_t = \mathbf{J}_{t-1} + \alpha_t^{\text{push}}(\mathbf{v}_t^{\text{push}} - \mathbf{J}_{t-1}\mathbf{p}_t)\mathbf{p}_t^\top \tag{5}$$

**3. Pop:** Similarly, to pop the top element of $\mathbf{J}_{t-1}$, we apply a Delta Rule update on $\frac{1}{2} \|\mathbf{J}\mathbf{p}_{t-1}\|^2$, with step size $\alpha_t^{\text{pop}}$.

$$\mathbf{J}_t = \mathbf{J}_{t-1} - \alpha_t^{\text{pop}}\mathbf{J}_{t-1}\mathbf{p}_{t-1}\mathbf{p}_{t-1}^\top \tag{6}$$

### 4.2. Unified stack update

We derive a unified state update rule that models stack dynamics through a sequential composition of Push and Pop operations. At the beginning, $\mathbf{J}_{t=0} = 0$.

1. **Pop:** The previous state is first modified to erase information at the previous pointer position $\mathbf{p}_{t-1}$:

$$\mathbf{J}_t' = \mathbf{J}_{t-1}(\mathbf{I} - \alpha_t^{\text{pop}}\mathbf{p}_{t-1}\mathbf{p}_{t-1}^\top)$$

2. **Push:** The intermediate state is then updated at the current pointer position $\mathbf{p}_t$ with the new value $\mathbf{v}_t^{\text{push}}$:

$$\mathbf{J}_t = \mathbf{J}_t'(\mathbf{I} - \alpha_t^{\text{push}}\mathbf{p}_t\mathbf{p}_t^\top) + \alpha_t^{\text{push}}\mathbf{v}_t^{\text{push}}\mathbf{p}_t^\top$$

By substitution, we obtain the composed transition rule:

$$\mathbf{J}_t = \mathbf{J}_{t-1} \underbrace{(\mathbf{I} - \alpha_t^{\text{pop}}\mathbf{p}_{t-1}\mathbf{p}_{t-1}^\top)}_{\text{Pop-Erase}} \underbrace{(\mathbf{I} - \alpha_t^{\text{push}}\mathbf{p}_t\mathbf{p}_t^\top)}_{\text{Push-Erase}}$$
$$+ \alpha_t^{\text{push}}\mathbf{v}_t^{\text{push}}\mathbf{p}_t^\top \tag{7}$$

### 4.3. Coupling Stack Memory with Associative Memory

Our proposed architecture integrates the differentiable stack $\mathbf{J}_t$ seamlessly with the fast weight memory $\mathbf{S}_t$ of a standard DeltaNet. The interaction between these two memory systems is defined by how the stack's output modulates the associative memory's update rule.

#### 4.3.1. INPUT PROJECTIONS AND GATING

At each timestep $t$, the input $\mathbf{x}_t$ is projected into queries $\mathbf{q}_t$, keys $\mathbf{k}_t$, and values $\mathbf{v}_t$ as in standard linear attention. Additionally, we project $\mathbf{x}_t$ to determining the stack action probabilities:

$$\alpha_t = \text{softmax}(\mathbf{W}_\alpha \mathbf{x}_t) \in \Delta^3$$

where $\alpha_t^{\text{push}}$ and $\alpha_t^{\text{pop}}$ correspond to the first two components of this distribution. This introduces minimal parameter overhead ($\mathbf{W}_\alpha \in \mathbb{R}^{d \times 3 \cdot H}$, where $H$ is the number of heads), making the stack control mechanism lightweight.

For the stack value update, we set $\mathbf{v}_t^{\text{push}} = \mathbf{v}_t$, reusing the standard value projection. This design enforces a **shared representation space**, ensuring that the stack and the associative memory store distinct structural views of the same semantic content, and maintains **parameter efficiency**, as the stack introduces no additional parameters for value generation.

#### 4.3.2. STACK-AUGMENTED ASSOCIATIVE RECALL

The core innovation of DeltaStack lies in how the stack augments the associative memory $\mathbf{S}_t$.

We treat $\mathbf{v}_{t-1}^{\text{push}}$ as a prediction bias or context signal for DeltaNet update rule (Eq. 1). The associative memory update is then derived from the augmented online loss:

$$\mathcal{L}_{\text{aug}}(\mathbf{S}) = \frac{1}{2}\|(\mathbf{S}\mathbf{k}_t + \mathbf{v}_{t-1}^{\text{read}}) - \mathbf{v}_t\|^2$$

This implies that the associative memory $\mathbf{S}_t$ only needs to learn the *residual* error between the target value $\mathbf{v}_t$ and the stack's prediction $\mathbf{v}_{t-1}^{\text{read}}$. If the stack correctly predicts the next value (e.g., in a Dyck language task where the closing bracket is deterministic given the stack top), the associative memory update becomes zero, effectively offloading the task to the stack.

The resulting update rule for the associative memory $\mathbf{S}_t$ becomes:

$$\mathbf{S}_t = \mathbf{S}_{t-1} - \beta_t(\mathbf{S}_{t-1}\mathbf{k}_t + \mathbf{v}_{t-1}^{\text{read}} - \mathbf{v}_t)\mathbf{k}_t^\top$$

where $\beta_t$ is the dynamic step size. By rearranging the terms, we recover a DPLR transition structure similar to DeltaNet:

$$\mathbf{S}_t = \mathbf{S}_{t-1}\left(\mathbf{I} - \beta_t\mathbf{k}_t\mathbf{k}_t^\top\right) + \beta_t\left(\mathbf{v}_t - \mathbf{v}_{t-1}^{\text{read}}\right)\mathbf{k}_t^\top \quad (8)$$

*Remark* 4.1. Since $\mathbf{J}_t$ gets updated (Eq. 7) independently from $\mathbf{S}_t$, we could in principle equip $\mathbf{J}_t$ to other linear attention methods such as RWKV-7 (Peng et al., 2025), TTT-Linear (Sun et al., 2025) or Gated DeltaNet (Yang et al., 2025). This structural independence establishes our stack mechanism as a general-purpose module, capable of augmenting the expressivity of the broader family of linear attention architectures without requiring specific modifications to their core recurrence dynamics.

*Remark* 4.2. Despite the sequential nature of stack operations, our formulation remains fully parallelizable. The stack update rule derived in Section 4.2 (Eq. 7) is a composition of rank-1 updates. By expanding the time dimension to explicitly interleave "Pop" and "Push" steps (mapping $T$ steps to $2T$ virtual steps), Eq. 7 can be computed using DeltaNet's chunkwise parallel algorithm. Similar, by setting $\mathbf{v}_t$ in Eq. 1 equal to $\mathbf{v}_t - \mathbf{v}_{t-1}^{\text{read}}$, we can use the same chunkwise parallel algorithm for computing Eq. 8. Overall, the total complexity is $O(LCd + Ld^2)$.

## 4.4. Theory

### 4.4.1. STATE-TRACKING PROBLEMS

It is worth noting that DeltaStack generalizes the original DeltaNet architecture. Specifically, DeltaNet can be recovered as a special case of DeltaStack where the stack mechanism is effectively disabled. Given that the input $\mathbf{x}_t$ typically undergoes normalization, the projection layer can effectively silence the push gate by setting the weights $\mathbf{W}_\alpha$ to zero and assigning a sufficiently large negative value to the bias term associated with the PUSH action relative

to the NO-OP bias. Under this configuration, the stack remains empty ($\mathbf{J}_t = \mathbf{0}$), the read term $\mathbf{v}_{t-1}^{\text{read}}$ vanishes, and we recover the associative memory update rule of standard DeltaNet (Eq. 1).

Consequently, DeltaStack can recover the state-tracking expressive power of DeltaNet, as formalized by the following Proposition.

**Proposition 4.3.** *A one-layer DeltaStack with finite precision can solve an* $\mathsf{NC}^1$*-complete problem under* $\mathsf{AC}^0$*-reductions.*

*Proof.* Theorem 3 from Grazzi et al. (2025) showed that the DPLR form of DeltaNet's state transition matrix (with $\beta_t \in [0, 2]$) allows it to solve the $\mathsf{NC}^1$-complete 5-element state tracking problem. Since DeltaStack can recover DeltaNet as a special case, it follows that DeltaStack can solve this same $\mathsf{NC}^1$-complete problem. $\qquad\square$

### 4.4.2. DYCK LANGUAGE

We show that a single-layer DeltaStack can recognize the $\text{Dyck}_{k,D}$ language, ie. the language of balanced parentheses consisting of $k$ distinct bracket types and the nesting depth never exceeds $D$ (Hewitt et al., 2020; Yao et al., 2021).

**Theorem 4.4.** *A single-layer DeltaStack model with embedding dimension* $d = \lceil \log_2(k) \rceil + 2$ *and stack capacity* $S \geq D$ *can recognize the language Dyck$_{k,D}$.*

*Remark* 4.5. For detailed proof, we refer readers to the Appendix A. Theorem 4.4 highlights several key advantages of DeltaStack over existing expressivity results of $\text{Dyck}_{k,D}$ language for Recurrent Neural Networks (RNNs), State Space Models (SSMs), and Transformers:

- **Parameter Efficiency vs. RNNs:** While standard RNNs can recognize $\text{Dyck}_{k,D}$ using a hidden state size of $O(D \log k)$ (Hewitt et al., 2020), their weight matrices typically scale with the state size, making the parameter count dependent on the maximum depth $D$. In contrast, our construction utilizes the stack memory $\mathbf{S}_t$ to handle depth, keeping the model's learnable parameters (matrices $\mathbf{W}_Q, \mathbf{W}_K, \mathbf{W}_V$) independent of $D$. This decoupling allows DeltaStack to generalize to deeper nesting at test time simply by increasing the stack capacity $S$, without retraining.

- **Layer Efficiency vs. Transformers and SSMs:** Previous expressivity results for recognizing Dyck languages often require multi-layer architectures. For instance, Transformers typically require $O(D)$ layers (depth-wise recursion) to model nesting depth $D$ via self-attention (Yao et al., 2021), and Mamba/SSMs require at least 2 layers to implement the necessary gating logic (Theorem 6 in (Sarrof et al., 2024)). Our result

demonstrates that DeltaStack can solve $\text{Dyck}_{k,D}$ with a **single layer**, attributable to the explicit read/write operations of the differentiable stack mechanism.

## 5. Experiments

### 5.1. Formal languages

We adopt the experimental framework in (Walker et al., 2025) for formal language tasks, categorizing them via the Chomsky hierarchy (Deletang et al., 2023). We compare our approach against parallelizable linear attention baselines such as DeltaNet (Yang et al., 2024b), Gated DeltaNet (Yang et al., 2025), Gated DeltaProduct (Siems et al., 2025), RWKV-7 (Peng et al., 2025), Transformer (Vaswani et al., 2017) and Stack Attention (DuSell & Chiang, 2024). We also include non-linear recurrent baselines such as LSTM (Hochreiter & Schmidhuber, 1997) and Stack-RNN (Joulin & Mikolov, 2015) as references for ideal state-tracking performance. Please refer to the Appendix B for further details on task specifications, training procedures, and model architectures.

Table 1 reports the results. DeltaStack significantly outperforms all other parallel baselines (Group 2) across both Regular and DCF tasks, achieving an average accuracy of 79.3% compared to 65.5% for the strongest baseline (DeltaNet). Notably, on structure-heavy DCF tasks such as *Stack Manipulation* and *Dyck*, DeltaStack nearly matches the performance of the sequential Stack-RNN, validating that our linearized stack mechanism successfully bridges the gap between parallel efficiency and hierarchical expressivity.

### 5.2. Synthetic Associative Recall Tasks

To verify that the strong inductive bias of the LIFO stack does not degrade the base model's retrieval capabilities, we evaluate DeltaStack on the Mechanistic Architecture Design (MAD) benchmark (Poli et al., 2024). As shown in Table 2, DeltaStack achieves an average score of 72.2, slightly surpassing the DeltaNet baseline (71.8). Crucially, it matches DeltaNet's perfect performance (100.0) on pure retrieval tasks like In-Context Recall and Selective Copy, confirming that the stack integration does not limit the model's in-context retrieval capabilities.

### 5.3. Language modeling

**Standard LM Benchmarks.** To evaluate general language capabilities, we pretrained DeltaStack and our primary baseline, DeltaNet, at 340M (15B tokens) and 760M (30B tokens) parameter scales on FineWeb-Edu (Penedo et al., 2024). All models were trained identically (4096 context length, 0.5M token global batch size; see Appendix B). We evaluated WikiText perplexity and zero-shot commonsense

reasoning (LAMBADA (Penedo et al., 2024), PIQA (Bisk et al., 2020), HellaSwag (Zellers et al., 2019), WinoGrande (Sakaguchi et al., 2021), ARC (Clark et al., 2018)) using `lm-evaluation-harness`.

As shown in Table 3, DeltaStack matches DeltaNet's performance at the 340M scale across both tape configurations ($s = 8, 16$). This confirms that integrating the differentiable spatial tape fully preserves the core modeling capacity of the linear attention backbone. Furthermore, when scaled to 760M parameters, DeltaStack ($s = 8$) consistently improves perplexity and achieves a higher average zero-shot accuracy (52.0 vs. 51.4) compared to DeltaNet, validating its architectural benefits and scalability for larger-scale modeling.

**Length extrapolation.** Figure 1 shows results on 2 long-context corpora: CodeParrot for coding and OpenThoughts-114k-Math (Team, 2025) for math. To evaluate the robustness of DeltaStack on long-context reasoning, we evaluated our 340M-parameter models on sequence lengths up to 16k tokens, on **CodeParrot** (coding) and **OpenThoughts Math** (mathematical reasoning) benchmarks. Figure 1 presents the token-wise loss across extended sequence lengths. Both DeltaStack variants consistently achieve lower per-token loss than the DeltaNet baseline across the entire 16k context, although all three models' performance degrades considerably on OpenThoughts-114k-Math. Crucially, the performance gap is maintained even as the models extrapolate far beyond the training window.

### 5.4. Analysis

**Effect of stack size**: Figure 3 illustrates the impact of stack size ($S \in \{8, 16, 24, 48\}$) on model accuracy across three formal language tasks: PARITY, REVERSE STRING, and STACK MANIPULATION. We follow the experiment settings of Section 5.1 for this analysis. We observe two primary trends. First, increasing the stack capacity significantly improves the performance on tasks requiring hierarchical memory, which is consistent with theoretical bounds established in prior work (Hewitt et al., 2020). Second, the model maintains perfect state-tracking capabilities regardless of stack size.

**Visualization of Stack Dynamics.** We evaluated a trained DeltaStack model on the Dyck-$k$ task and recorded the pointer trajectory $p_t$ for each attention head, under the same training and testing settings in Section 5.1. We then measured the Pearson correlation coefficient ($r$) between the pointer's value and the ground-truth nesting depth $d_t$ specifically during the predictive phase (i.e., the generation of closing brackets). Figure 4 displays the trajectory of the primary structural head (Layer 1, Head 3) alongside the ground-truth depth. The visualization reveals a near-perfect alignment ($r > 0.99$) and confirms that DeltaStack success-

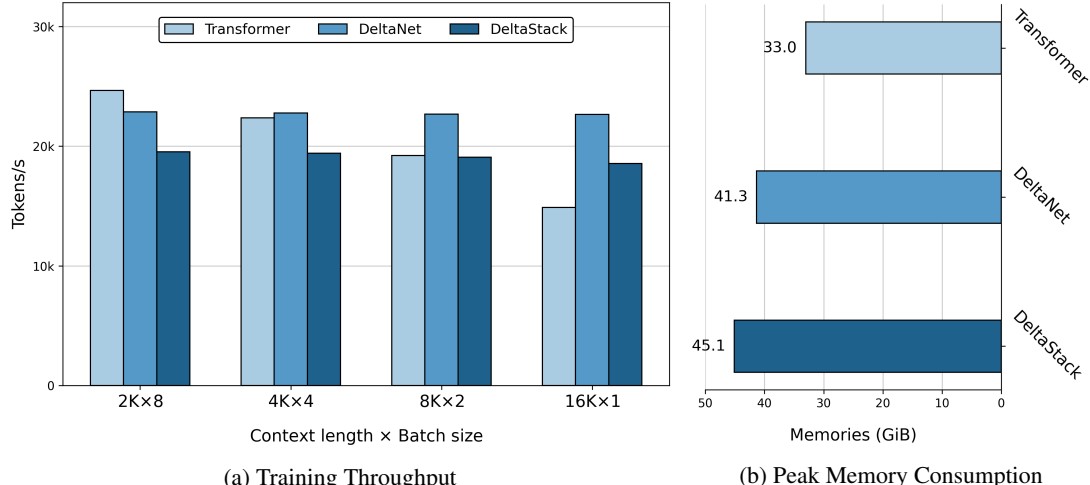

(a) Training Throughput

(b) Peak Memory Consumption

*Figure 2.* **Training Efficiency Metrics.** Results for 1.3B parameter models on a single A100 GPU. (a) Training throughput (tokens/s) across varying context lengths and batch sizes (maintaining constant total tokens). (b) Peak memory consumption (GiB) during training.

*Table 1.* **Results for formal language tasks.** Average and standard deviation of validation accuracy over five runs. Most baseline results in 'Regular' tasks were taken from Walker et al. (2025). DeltaStack consistently outperforms all parallel baselines (Group 2) and approaches the performance of sequential baselines (Group 1) on hierarchical tasks.

| Model | Regular (R) | | | | DCF | | | | |
| | Cycle Nav. | Even Pairs | Mod Arith. No Brack. | Parity | Stack Manipulation | Reverse String | Dyck | Mod Arith. w/ Brack. | Average |
|---|---|---|---|---|---|---|---|---|---|
| LSTM | $100.0 \pm 0.0$ | $100.0 \pm 0.0$ | $99.9 \pm 0.1$ | $100.0 \pm 0.0$ | $76.7 \pm 1.8$ | $62.3 \pm 0.9$ | $61.2 \pm 1.4$ | $61.6 \pm 2.6$ | 82.7 |
| Stack-RNN | $96.3 \pm 2.7$ | $100.0 \pm 0.0$ | $100.0 \pm 0.0$ | $98.3 \pm 0.5$ | $98.5 \pm 1.8$ | $99.2 \pm 0.2$ | $98.9 \pm 0.8$ | $79.4 \pm 3.5$ | 96.3 |
| DeltaNet | $46.7 \pm 6.1$ | $100.0 \pm 0.0$ | $66.4 \pm 8.8$ | $97.7 \pm 2.0$ | $64.3 \pm 1.2$ | $56.6 \pm 1.5$ | $58.7 \pm 2.5$ | $33.2 \pm 1.2$ | 65.5 |
| Gated DeltaNet | $53.8 \pm 8.8$ | $100.0 \pm 0.0$ | $42.8 \pm 8.2$ | $56.5 \pm 1.9$ | $63.7 \pm 2.8$ | $58.2 \pm 2.0$ | $57.3 \pm 2.3$ | $34.3 \pm 1.4$ | 58.3 |
| Gated DeltaProduct | $46.3 \pm 6.6$ | $100.0 \pm 0.0$ | $78.4 \pm 10.9$ | $98.0 \pm 1.4$ | $54.9 \pm 2.4$ | $58.7 \pm 1.3$ | $57.0 \pm 0.7$ | $34.8 \pm 1.5$ | 66.0 |
| RWKV-7 | $37.8 \pm 5.0$ | $88.1 \pm 14.2$ | $39.5 \pm 6.1$ | $51.1 \pm 0.3$ | $55.1 \pm 1.9$ | $57.7 \pm 1.2$ | $57.3 \pm 0.8$ | $30.9 \pm 0.9$ | 52.2 |
| Transformer | $24.4 \pm 0.5$ | $90.4 \pm 10.4$ | $23.6 \pm 0.7$ | $52.2 \pm 0.4$ | $57.1 \pm 1.9$ | $58.8 \pm 1.5$ | $52.5 \pm 1.6$ | $27.2 \pm 1.7$ | 48.3 |
| Stack-Attention | $24.7 \pm 1.3$ | $97.1 \pm 2.8$ | $29.5 \pm 0.8$ | $51.3 \pm 0.5$ | $57.4 \pm 1.2$ | $88.3 \pm 1.1$ | $88.4 \pm 2.9$ | $30.1 \pm 2.2$ | 58.4 |
| **DeltaStack** | $46.4 \pm 5.7$ | $100.0 \pm 0.0$ | $65.4 \pm 7.7$ | $98.1 \pm 0.4$ | $96.8 \pm 2.9$ | $98.8 \pm 0.8$ | $94.3 \pm 3.4$ | $35.2 \pm 1.9$ | 79.4 |

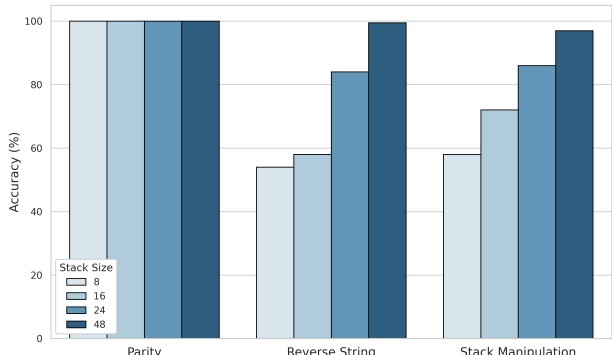

*Figure 3.* Accuracy vs Stack Size for Formal Language Tasks

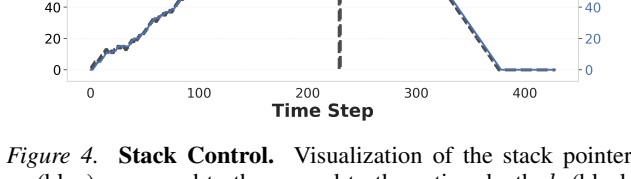

*Figure 4.* **Stack Control.** Visualization of the stack pointer $p_t$ (blue) compared to the ground-truth nesting depth $d_t$ (black dashed) on a Dyck-$k$ sequence.

fully learns to operate an explicit, interpretable stack data structure to solve recursive tasks.

**Efficiency** We evaluate 1.3B parameter models on a single A100 GPU, comparing DeltaStack against DeltaNet and a

standard Transformer. As shown in Figure 2(a), DeltaStack incurs a modest memory overhead (45.1 GiB) compared to DeltaNet (41.3 GiB) and the Transformer (33.0 GiB) due to its stack mechanism. However, it exhibits superior length-scaling properties (Figure 2(b)). While the Transformer's

*Table 2.* **Results on the MAD benchmark.** Baselines' results are borrowed from Yang et al. (2024b).

| Model | Compress | Fuzzy Recall | In-Context Recall | Memorize | Noisy Recall | Selective Copy | Avg |
|---|---|---|---|---|---|---|---|
| Transformer | 51.6 | 29.8 | 94.1 | 85.2 | 86.8 | 99.6 | 74.5 |
| Mamba | 52.7 | 6.7 | 90.4 | 89.5 | 90.1 | 86.3 | 69.3 |
| GLA | 38.8 | 6.9 | 80.8 | 63.3 | 81.6 | 88.6 | 60.0 |
| DeltaNet | 42.2 | 35.7 | 100 | 52.8 | 100 | 100 | 71.8 |
| **DeltaStack** | 43.4 | 36.2 | 100 | 53.2 | 100 | 100 | 72.2 |

*Table 3.* Results on perplexity and zero-shot commonsense reasoning tasks evaluated at the 340M and 760M parameter scales.

| Model | Wiki. ppl $\downarrow$ | LMB. ppl $\downarrow$ | LMB. acc $\uparrow$ | PIQA acc $\uparrow$ | Hella. acc_n $\uparrow$ | Wino. acc $\uparrow$ | ARC-e acc $\uparrow$ | ARC-c acc_n $\uparrow$ | Avg. |
|---|---|---|---|---|---|---|---|---|---|
| | | | *340M params / 15B tokens* | | | | | | |
| DeltaNet | 28.22 | 30.22 | 33.4 | 65.9 | 38.9 | 51.1 | 56.9 | 26.9 | 45.5 |
| **DeltaStack** ($s = 8$) | 28.15 | 30.15 | 33.9 | 66.8 | 39.5 | 50.7 | 57.5 | 27.1 | 45.9 |
| **DeltaStack** ($s = 16$) | 27.97 | 30.26 | 34.4 | 66.1 | 39.8 | 50.9 | 57.8 | 28.3 | 46.2 |
| | | | *760M params / 30B tokens* | | | | | | |
| DeltaNet | 21.89 | 19.47 | 38.9 | 69.5 | 47.2 | 56.4 | 65.1 | 31.5 | 51.4 |
| **DeltaStack** ($s = 8$) | 21.81 | 18.26 | 41.4 | 70.7 | 47.6 | 55.2 | 65.3 | 31.6 | 52.0 |

throughput degrades severely from ∼25k to ∼15k tokens/s as context increases to 16K, DeltaStack maintains a highly stable rate of ∼19k tokens/s. Although slightly slower than DeltaNet (∼23k tokens/s), DeltaStack's linear complexity enables it to significantly outperform the standard Transformer in long-context regimes.

## 6. Discussion and Limitations

While DeltaStack successfully bridges the gap between the efficiency of linear attention and the structural expressivity of stack-based models, several limitations and avenues for future work remain.

**Theoretical Expressivity vs. Learnability.** Our theoretical analysis (Theorem 4.2) provides a constructive proof that DeltaStack is *expressive* enough to recognize Dyck-$k$ languages. However, expressivity does not guarantee learnability, and DeltaStack still struggles with relatively complex DCF task such as "Modular Arithmetic with Brackets" (Table 1). This suggests that while the architecture admits a solution, the optimization landscape may be difficult to traverse without additional inductive biases.

**Scale and Complexity.** Our empirical validation is currently limited to synthetic formal languages and small-scale language modeling (340M/760M parameters). While the length extrapolation results on CodeParrot (Figure 1) are promising, it remains an open question how the differentiable stack interacts with the massive semantic knowledge

in larger models.

**Retrieval Limitations and Hybrid Architectures.** Despite the improved expressivity on formal languages, DeltaStack remains a linear attention model at its core, and hence underperforms Transformer on long-context associative recall task (Arora et al., 2024). A promising path forward is a hybrid architecture that combines the efficient inference and expressiveness of DeltaStack with the precise retrieval capabilities of Transformers.

## 7. Conclusion

In this work, we introduced DeltaStack, a novel architecture that bridges the gap between the computational efficiency of linear attention and the structural expressivity of stack-augmented models. Our theoretical analysis confirms that DeltaStack extends the expressivity of DeltaNet to recognize Dyck-$k$ languages, while empirical results demonstrate superior performance on formal language benchmarks and robust length generalization on code generation tasks. Future work offers several promising directions. Scaling up DeltaStack to billion-parameter models will be critical to understanding how stacks interact with massive semantic knowledge. Second, theoretical efforts could focus on extending our expressivity guarantees to the full class of DCFLs. Furthermore, investigating hybrid architectures that combine DeltaStack with softmax attention could further unlock the potential of hierarchical reasoning in conjunction with long-context retrieval.

## Acknowledgments

The work of Anh T Nguyen, Saleh Momeni and Bing Liu was supported in part by two NSF grants (IIS-2229876 and CNS-2225427), and an NVIDIA's Academia Grant, which provides cloud compute via its Saturn Cloud.

## Impact Statement

This paper presents work whose goal is to advance the field of Machine Learning. There are many potential societal consequences of our work, none which we feel must be specifically highlighted here.

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

# A. Proof for Theorem 4.4

## A.1. Preliminaries: The Dominant Error Lemma

To guarantee that multiple errors do not sum to zero (cancel out) within $\mathbf{S_t}$, we establish the following lemma.

**Lemma A.1.** *Let $\{\delta_1, \ldots, \delta_N\}$ be a sequence of error vectors in $\mathbb{Z}^d$ where any non-zero $\delta_i$ satisfies $\|\delta_i\|_\infty \geq 1$. Let $\gamma \in (0, 1/2)$ be a decay factor. Consider the accumulated sum $H_N = \sum_{i=1}^{N} \gamma^{N-i}\delta_i$. If there exists any $i$ such that $\delta_i \neq \mathbf{0}$, then $H_N \neq \mathbf{0}$.*

*Proof.* Let $\tau$ be the index of the *last* non-zero error (i.e., $\delta_\tau \neq \mathbf{0}$). We isolate the term at $\tau$ and bound the sum of all preceding errors (the "tail") using the geometric series. For a dimension $m$ where $|\delta_\tau^{(m)}| \geq 1$:

$$\left| \sum_{j=1}^{\tau-1} \gamma^{\tau-j}\delta_j^{(m)} \right| \leq \sum_{k=1}^{\infty} \gamma^k = \frac{\gamma}{1-\gamma}$$

For the total sum to be non-zero, the magnitude of the last error $\delta_\tau$ must strictly exceed the "tail". This holds when $1 > \frac{\gamma}{1-\gamma} \iff \gamma < 1/2$. $\qquad\square$

## A.2. Constructive Proof

We restate Theorem 4.4 here:

**Theorem A.2.** *A single-layer DeltaStack model with embedding dimension $d = \lceil \log_2(k) \rceil + 2$ and stack capacity $S \geq D$ can recognize the language $Dyck_{k,D}$. Specifically, there exists a parameterization such that the final memory readout $\|\mathbf{o}_T\| = 0$ if and only if the input sequence is in $Dyck_{k,D}$.*

**Representation** To satisfy the memory bound $d = O(\log k)$, we assign a unique binary vector $\mathbf{u}_c \in \{0,1\}^d$ to each bracket type $c \in \{1, \ldots, k\}$, plus distinct vectors for a "Guard" token $\mathbf{u}_G$ and "EOS" token. Crucially, for any distinct types $i \neq j$, the difference vector $\mathbf{u}_i - \mathbf{u}_j$ contains at least one non-zero integer component.

**Projections and Logic** We map opposite brackets to the **same** embedding code:

$$\mathbf{v}_t = \mathbf{W}_V\mathbf{x}_t = \begin{cases} \mathbf{u}_i & \text{if } x_t = (_i \text{ or } )_i \\ \mathbf{u}_G & \text{if } x_t = \text{EOS} \end{cases}$$

We fix a constant key and query $\mathbf{k}_t = \mathbf{q}_t = \mathbf{k}$ (where $\|\mathbf{k}\| = 1$) and define the action gating $\mathbf{W}_\alpha$ as follows:

1. **Open Bracket** $(_i$: Action is **Push**, i.e. we set $\alpha_t^{\text{push}} = 1$. The step size is set to $\beta_t = 0$, $\mathbf{J}_t$ gets populated with $\mathbf{u}_i$ and $\mathbf{S}_t$ remains unchanged. The condition $S \geq D$ ensures that $\mathbf{J}_t$ never overflows, i.e. it can store $D$ 'non-overlapping' embedding vectors.

2. **Check Step** $(_i$ or **EOS**): Action is **Pop**, i.e. we set $\alpha_t^{\text{pop}} = 1$. The step size is set to $\beta_t = \beta \in (1/2, 1.0)$, implying a decay factor $\gamma = (1 - \beta) < 1/2$.

**Dynamics** Given the update rule in Eq. 8, and initializing $\mathbf{S}_0 = \mathbf{0}$, the matrix $\mathbf{S}_t$ has the form $\mathbf{h}_t\mathbf{k}^\top$. The history vector $\mathbf{h}_t$ evolves as:

$$\mathbf{h}_t = (1 - \beta_t)\mathbf{h}_{t-1} - \beta_t(\underbrace{\mathbf{v}_{t-1}^{\text{read}} - \mathbf{v}_t}_{\Delta_t}) \tag{9}$$

During **check steps**, this becomes a decay-and-add recurrence: $\mathbf{h}_t = \gamma\mathbf{h}_{t-1} - \beta\Delta_t$

**Correctness** We analyze the residual $\Delta_t$ during **check steps**:

- **Valid Match:** The input is $)_i$ and the stack top is $(_i$. Since both map to $\mathbf{u}_i$, $\Delta_t = \mathbf{u}_i - \mathbf{u}_i = \mathbf{0}$.

- **Structure Mismatch:** The current token is $)_i$ but the stack top is $(_j$. $\Delta_t = \mathbf{u}_j - \mathbf{u}_i \neq \mathbf{0}$.

- **Boundary Violation:** The stack is empty and the current token is $)_i$ ($\mathbf{v}_{t-1}^{\text{read}} = \mathbf{u}_G$, $\mathbf{v}_t = \mathbf{u}_i$), or non-empty stack at EOS ($\mathbf{v}_{t-1}^{\text{read}} = \mathbf{u}_i$, $\mathbf{v}_t = \mathbf{u}_G$). In both cases, $\Delta_t \neq \mathbf{0}$.

At the end of sequence $T$, $\mathbf{h}_T = \beta \sum_{j \in \text{Checks}} (1 - \beta)^{N_j} \Delta_j$, where $N_j$ is the number of check steps from time $h$ to $T$.

If the sequence is valid, all $\Delta_j = \mathbf{0}$, thus $\mathbf{h}_T = \mathbf{0}$ by Eq. 9.

If invalid, there exists some $\Delta_j \neq \mathbf{0}$. By Lemma A.1, the fast decay ensures $\mathbf{h}_T \neq \mathbf{0}$.

A simple norm-based threshold by the MLP layer on $\mathbf{o}_T = \mathbf{S}_T \mathbf{q}_t = \mathbf{h}_T \mathbf{k}^\top \mathbf{k} = \mathbf{h}_T$ to check whether $\mathbf{h}_T = 0$ ensures the validity of the whole input string. Hence, we have derived a parameterization of DeltaStack such that the final memory readout $\|\mathbf{o}_T\| = 0$ if and only if the input sequence is valid, recognizing the language $\text{Dyck}_{k,D}$. $\qquad\square$

# B. Experiment details

## B.1. Formal Languages

### B.1.1. TASK DESCRIPTION

We adopt the formal language benchmarks from Deletang et al. (2023), specifically the 4 regular language tasks (*Even Pairs*, *Modular Arithmetic (simple)*, *Parity Check*, and *Cycle Navigation*) and 4 Deterministic Context-Free (DCF) tasks (*Stack Manipulation*, *Reverse String*, *Dyck*, and *Modular Arithmetic with Brackets*). We introduce the *Dyck* task in addition to 7 other tasks from the original benchmark. We omit the DCF task *Solve Equation* as it is strictly harder than the *Modular Arithmetic* task, which all parallel models, including DeltaStack, already struggle with.

The tasks are defined as follows:

**Regular Languages.**

- **Even Pairs:** Given a binary string of $a$'s and $b$'s, determine if the total count of 'ab' and 'ba' substrings is even. This task requires tracking a simple 2-state transition system.

- **Modular Arithmetic (No Brackets):** Evaluate a linear arithmetic expression involving addition $(+)$, subtraction $(-)$, and multiplication $(\times)$ modulo 5 (e.g., $1 + 2 \times 4 \pmod 5$). The input contains digits $0 - 4$ and operators, processed sequentially without hierarchical precedence.

- **Parity Check:** Determine if the number of $b$'s in a sequence of $a$'s and $b$'s is even. This is the canonical regular language task equivalent to computing the sum modulo 2.

- **Cycle Navigation:** Navigate a directed cycle graph of length 5. The input is a sequence of actions ("move clockwise", "move counter-clockwise", "stay"), and the goal is to predict the final node index starting from the origin.

**Deterministic Context-Free Languages (DCF).**

- **Stack Manipulation:** Given a sequence of explicit `PUSH a`, `PUSH b`, or `POP` actions applied to an initially empty stack, predict the final stack content as if popping it from top to bottom. This task explicitly tests the model's ability to simulate stack dynamics.

- **Reverse String:** Given a binary string (e.g., `aabba`), generate its reverse (e.g., `abbaa`). This task requires the model to push the entire input onto a stack until a delimiter is reached, then pop to generate the output.

- **Dyck:** The input is a prefix of a valid balanced parentheses string involving multiple bracket types. The goal is to predict the sequence of valid closing parentheses required to complete the string. This task generalizes *Reverse String* by strictly enforcing well-nestedness without explicit pop commands.

- **Modular Arithmetic (w/ Brackets):** Evaluate an arithmetic expression modulo 5 that includes nested brackets (e.g., $-(1 - 2) \cdot (4 - 3)$). Unlike the regular version, this task requires a stack to handle the hierarchical precedence defined by the parentheses.

### B.1.2. HYPERPARAMETER SETTINGS

To ensure a rigorous comparison, we standardize the model configurations across DeltaStack and all baselines (DeltaNet, RWKV-7, Transformer, Stack-Attention, etc.), and describe the hyperprameters settings in Table 4. For DeltaStack, the stack size $s$ is set to 48 during training, and extended to 256 during testing, matching the evaluation protocol in Deletang et al. (2023).

*Table 4.* **Hyperparameters for Formal Language Tasks.**

| Hyperparameter | Value |
| --- | --- |
| Number of Layers | 2 |
| Hidden Dimension ($d_{\text{model}}$) | 128 |
| Optimizer | AdamW |
| Peak Learning Rate | $1 \times 10^{-3}$ |
| Min Learning Rate | $1 \times 10^{-5}$ |
| Weight Decay | 0.01 |
| Scheduler | Linear Warmup + Cosine Decay |
| Warmup Steps | 1,000 |
| Total Training Steps | 100,000 |
| Batch Size | 64 |
| Train Sequence Length | $[3, 40]$ (Uniformly sampled) |
| Eval Sequence Length | $[40, 256]$ |

## B.2. Language modeling

We adopt the experimental configuration from Yang et al. (2024b), utilizing the `flame` training pipeline from the `flash-linear-attention` repository for all language modeling experiments. All models were trained on a cluster of 8 H100 GPUs using the AdamW optimizer with a weight decay of 0.01. We employed a cosine annealing learning rate schedule with a peak learning rate of $1 \times 10^{-3}$ and a warmup period of 1,000 steps. To ensure stability, gradients were clipped at a maximum norm of 1.0. For the model architecture, we standardized the head dimension at 128 and set the kernel size for the short convolution layers to 4.

*Table 5.* **Hyperparameters for Language Modeling.**

| Hyperparameter | Value |
| --- | --- |
| Batch Size (Global) | 0.5M tokens |
| Context Length | 4096 |
| Optimizer | AdamW |
| Peak Learning Rate | $1 \times 10^{-3}$ |
| Weight Decay | 0.01 |
| Gradient Clipping | 1.0 |
| LR Scheduler | Cosine Annealing |
| Warmup Steps | 1,000 |
| Head Dimension ($d_{\text{head}}$) | 128 |
| Conv Kernel Size | 4 |
| Hardware | $8 \times$ NVIDIA H100 |

