# OpenReview forum: "Push, Pop, Parallelize: Stack-Augmented Linear Attention via the Delta Rule"
_ICML.cc/2026/Conference — ICML 2026 regular_

### Official Review · Reviewer_HDdX · 2026-03-07

**Soundness:** 3
**Presentation:** 4
**Significance:** 3
**Originality:** 3
**Overall Recommendation:** 4
**Confidence:** 3

**Summary:**

This paper presents DeltaStack - an extension of DeltaNet with a enhanced differentiable data store. Compared to DeltaNet, where the data store is a single memory matrix $S_t \in \mathbb{R}^{d \times d}$, DeltaStack adds additional memory $J_t \in \mathbb{R}^{s \times d}$ which can be interpreted as a linear sequence of $s$ vectors of dimension $d$ together with a scalar pointer $p_t \in [0, s-1]$. Note that the memory store is per-attention head.

The goal of the new method is to combine the efficiency of parallel training of linear transformers with the expresivity of stack-augmented architectures. Through extensive experiments, authors show that the new method improves performance on simple regular and deterministic context-free (DCF) languages without any degradation on a suite of standard language modeling tasks and an associative recall benchmark. Additionally, the method improves language modeling on two code and math related datasets, especially when the sequence length exceeds that encountered during training.

In formal languages, authors also provide theoretical results showing that while DeltaNet can recognize any regular language, DeltaStack can also recognize the $\text{Dyck}_{k,D}$ language.

**Compliance With Llm Reviewing Policy:**

Affirmed.

**Final Justification:**

My final score mirrors my initial evaluation and author's rebuttal. Based on the rebuttal, I believe the paper still needs certain edits before publication. However, I also recognize the paper's contribution, leading to my scoring leaning toward Accept.

**Key Questions For Authors:**

1. Can you justify your claim that natural language can be formalized as DCFL?

**Limitations:**

yes

**Strengths And Weaknesses:**

*Note on notation*: By [104l] or [048r], I mean line 104 left and line 48 right, respectively.

**Soundness:**

- Although the Abstract mentions a theoretical result proving that DeltaStack can model both regular and hierarchical languages [030l], only one specific (and arguably trivial) hierarchical language $\text{Dyck}_{k,D}$ is theoretically addressed (Theorem 4.4).
- Few points that would benefit from more justification:
	- [024r] "Natural language is not merely a linear sequence of tokens but a nested hierarchy of phrases and clauses—a structure formalized by Deterministic Context-Free Languages (DCFLs) (Chomsky, 1956)" - There is an implicit claim that improvements on DCFLs will lead to improvements in natural languages, motivating approaches such as DeltaStack. However, is natural language really context-free, and is it really deterministic? The reviewer believes that it is neither.
- Training hyperparameters, especially the learning rate, should be tuned for each method separately. This seems especially important across the split of recurrent vs parallel methods.


**Presentation:**

The exposition of the problem is clear. The architecture, training procedure, hyperparameter setting, and evaluation protocol are clearly stated.

Areas of improvement:
- The paper should address the potential wall time overhead of the newly introduced memory store and memory control, both during training and during inference. Now, only the asymptotic complexity is provided. One potential source of training overhead is that the $S_t$ update requires the stack prediction $v^\text{read}$, so the two operations cannot be performed in parallel. Another could be the expansion of the time dimension from $T$ steps to $2T$ virtual steps.
- Section 2 or 3 should briefly discuss the functioning of linear attention mechanisms, similarly to how DeltaNet is described in Section 3.2 (although possibly in less detail). Section 2 now only mentions that they "replace the softmax kernel with feature map approximations ϕ(·)" [088r], which does not contribute in understanding for an uninitiated reader. If saving space is needed, Section 4.2 could be shortened to reduce duplication with the preceding Section 4.1.
- It should be noted in the paper that the memory stores and pointers are per-attention head, which is only implicitly implied in [420l].

Some points would benefit from more explanation:
- (related to the previous point) [038r] "Bridging this gap requires equipping the model with unboundedly growing memory, typically in the form of a stack." - Is the choice of a stack really typical? Why is it the right choice?
- The curves in figure 1 seem to have been somehow smoothed, which should be mentioned in the text.

Very minor corrections:
- [132r] "stack-augmnented" -> "stack-augmented"
- Table 1 caption: "tasks from taken from" -> "tasks were taken from"
- Equation 7: missing space after the equation mark

**Significance:**

Quantitative improvements on the selected tasks are clearly demonstrated, especially in the area of simple formal languages. Qualitative experiments on the Dyck language in Section 5.4 show that the introduced memory-pointer approach behaves in a meaningful and interpretable manner in the given context. Thanks to its clear exposition, the paper can serve as a useful resource for the design of similar differentiable memory stores.

Still, it remains unclear whether the new method is directly applicable to tasks beyond formal languages. I would recommend exploring in more detail the promising performance of DeltaStack on code and math related benchmarks across extended sequence lengths. This might include comparison against baselines beyond DeltaStack (although not necessarily state-of-the-art baselines due to prohibitive compute demands) and evaluation of metrics beyond token loss (e.g. final answer accuracy in math problems).

**Originality:**

The paper clearly delimits its originality compared to existing work and combines existing approaches in a meaningfully original way.

---

> ### Author Rebuttal · Authors · 2026-03-29
>
> **Q:** Only one specific (and arguably trivial) hierarchical language is theoretically addressed, different from the claim of 'hierarchical languages' in the abstract.
>
> **A:** The $Dyck_{k,D}$ languages we analyze are the foundational languages of balanced brackets with $k$ types. Furthermore, despite the impressive success of modern LLMs, recent studies show they fail catastrophically at seemingly basic algorithmic tasks like parity tracking and matching parentheses [4, 5]. Rigorously studying these formal languages is crucial for robust reasoning [6].
>
> **Q:** Can natural language can be formalized as DCFL?
>
> **A:** Our phrasing in line 024 states that the nested hierarchical structures intrinsic to natural language (e.g., recursive grammar, clauses within clauses) can be formalized by DCFLs. Explicitly equipping a model to handle these hierarchical structures yields interpretable learned representations (as visualized in Figure 3) and provides robust length generalization that baseline models lack (as shown in Table 1).
>
> **Q:** Learning rate should be tuned for each method separately in Table 1.
>
> **A:** We observed that given the relative simplicity of the formal language datasets, all capable baseline models converged smoothly within 100k steps across a wide band of learning rates. We therefore selected the most stable learning rate ($1\times10^{-3}$).
> More importantly, most baselines provably cannot length-generalize on deep recursive tasks regardless of how extensively their learning rates are tuned.
>
> **Q:** The paper should address the potential wall time overhead.
>
> **A:** We perform experiments to demonstrate the training throughput ([link](https://ibb.co/21dDjB6x)), memory cost ([link](https://ibb.co/tTZwTvyH)) and inference latency ([link](https://ibb.co/Nd9vcHNS)) of 1.3B parameter models on a single A100 GPU. DeltaStack maintains competitive scaling at long contexts, though it currently exhibits a moderate throughput drop and memory increase compared to DeltaNet.
>
> **Q:** Missing linear attention.
>
> **A:** Standard softmax attention computes the output $o_t$ as a weighted sum of values $v_i$, where the weights are determined by softmax of the dot products between queries $q_t$ and keys $k_i$:
> $$o_t = \frac{\sum_{i=1}^t \exp(k_i^\top q_t) v_i}{\sum_{j=1}^t \exp(k_j^\top q_t)}$$
> Linear attention replaces the exponential kernel with a dot product of feature maps, $\phi(k_i)^\top \phi(q_t)$. In practice, it is common to drop the normalization denominator and use the identity function for the feature map.
>
> $$S_t = S_{t-1} + v_t k_t^\top, \quad o_t = S_t q_t$$
>
> **Q:** It should be noted that the memory stores and pointers are per-attention head.
>
> **A:** We completely agree. Similar to standard DeltaNet, all associative memory matrices ($S_t$) and stack matrices ($J_t$) are per attention head.
>
> **Q:** [038r] Is the choice of a stack really typical?
>
> **A:** The phrasing "typically in the form of a stack" describes the transition from Finite State Automata (which cannot model unbounded recursion) to Pushdown Automata. A stack is the standard external memory structure required to parse DCFL.
>
> **Q:** The curves in figure 1 seem to have been somehow smoothed.
>
> **A:** These curves are smoothed using a window of 500 tokens.
>
> **Q:** Minor writing corrections.
>
> **A:** We thank the reviewer for their careful reading and corrections. We will incorporate all of these edits into the final manuscript.
>
> **Q:** Exploring in more detail the promising performance of DeltaStack on code and math related benchmarks on metrics beyond token loss
>
> **A:** We agree that evaluating downstream generation metrics-such as exact match accuracy for OpenThoughts Math or pass@1 for CodeParrot-would provide a more comprehensive view of the model's actual reasoning capabilities. Due to the limited computational budget and time constraints of the rebuttal period, we were unable to run full autoregressive generation evaluations for larger scale models. Furthermore, achieving meaningful exact match accuracy or pass rates at our current model scales (340M and 760M parameters) is highly challenging, as these complex tasks typically require extensive semantic knowledge. Future work will explore utilizing post-training methods to effectively scale the proposed architecture to these more challenging coding and math tasks.
>
>
> [1] Hewitt, J., et al., "RNNs can generate bounded hierarchical languages with optimal memory,"
>
> [2] Shunyu Yao, et al., "Self-Attention Networks Can Process Bounded Hierarchical Languages,"
>
> [3] Yash Sarrof, et al. "The Expressive Capacity of State Space Models: A Formal Language Perspective."
>
> [4] William Merrill, et al. "The Illusion of State in State-Space Models."
>
> [5] Daking Rai, et al., "Failure by Interference: Language Models Make Balanced Parentheses Errors When Faulty Mechanisms Overshadow Sound Ones,"
>
> [6] Hu, M., et al., "Between Circuits and Chomsky: Pre-pretraining on Formal Languages Imparts Linguistic Biases,"

---

> > ### Author Rebuttal · Reviewer_HDdX · 2026-04-02
> >
> > Thank you for your detailed rebuttal and for your contribution, as well as for the additional performance measurements.
> >
> > I still believe that points 1 and 2 should be addressed in more detail. Although $Dyck_{k,D}$ is a standard benchmark in hierarchical languages, it's unclear to what extend it is representative. In [024r], the implicit claim is that natural language can be formalized by DCFL, which should be justified.

---

> > > ### Author Response · Authors · 2026-04-02
> > >
> > > We thank the reviewer for acknowledging our response and for their continued valuable feedback.
> > >
> > > **Q**: It is unclear to what extent the Dyck language is representative of DCFL languages.
> > >
> > > **A**: We agree that Dyck does not encompass the full complexity of all DCFLs. However, it serves as a canonical and well-studied instance of context-free structure. Because it isolates the mechanics of recursion (matching nested pairs), it is crucial for evaluating a model's capacity to handle simple hierarchical structures before scaling to more complex DCFLs [1, 2, 3].
> > >
> > > **Q**: In [024r], the implicit claim is that natural language can be formalized by DCFL, which should be justified.
> > >
> > > **A**: We agree that our phrasing on line [024r] - "a structure formalized by Deterministic Context-Free Languages" - is imprecise and could be misinterpreted as claiming that natural language in its entirety is a DCFL, which is overly reductive.
> > > Our intent was to highlight that DCFLs provide a useful formal framework for modeling certain recursive and hierarchical syntactic patterns (e.g., nested dependencies) that arise in natural language. We will revise this sentence in the final manuscript to explicitly clarify that while human language is predominantly semantically driven, modeling its recursive syntactic properties benefits from DCFL-like structural tracking.
> > >
> > > Ultimately, our goal is not to claim that DeltaStack can achieve state-of-the-art language modeling performance, but rather to demonstrate that DeltaStack can formally model a critical subset of DCFLs, maintain language modeling performance comparable to DeltaNet, and leverage its structural capacity to outperform DeltaNet on zero-shot length extrapolation in highly structured domains like code and math.
> > >
> > > [1] Yao, S., et al. "Self-attention networks can process bounded hierarchical languages,"
> > >
> > > [2] Y. Sarrof, Y. Veitsman, M. Hahn. "The expressive capacity of state space models: A formal language perspective,"
> > >
> > > [3] Hewitt, J., et al., "RNNs can generate bounded hierarchical languages with optimal memory,"

---

### Official Review · Reviewer_1Rzt · 2026-03-08

**Soundness:** 3
**Presentation:** 2
**Significance:** 2
**Originality:** 3
**Overall Recommendation:** 4
**Confidence:** 4

**Summary:**

This paper introduces DeltaStack, a new variant of DeltaNet that augments the original architecture with an explicit differentiable stack memory. The added stack memory is intended to improve performance on tasks that exhibit stack-like structure, particularly Deterministic Context-Free Language tasks such as Dyck languages. The paper evaluates DeltaStack on a range of language modeling and formal language tasks, and the results suggest that it outperforms the original DeltaNet as well as several other linear recurrent baselines.

**Compliance With Llm Reviewing Policy:**

Affirmed.

**Final Justification:**

The rebuttal response addresses most of my concerns. Due to the method utilizing more computation and memory, which may be a limitation, and my original score is a positive one, I have decided to maintain my score.

**Key Questions For Authors:**

See weaknesses

**Limitations:**

Yes

**Strengths And Weaknesses:**

# Strengths

- The central idea of augmenting a linear recurrent model with stack memory is novel and well motivated. The argument that stack-like memory may help language models on hierarchical or algorithmic tasks is promising.

- The paper is generally clear and well structured, and the experimental section provides reasonable support for the main claims.

- The evaluation covers multiple tasks and includes comparisons against several relevant baselines, which strengthens the empirical case for the proposed method.

# Weaknesses

- The paper does not clearly explain how the output of the DeltaStack token mixer is computed. In particular, the interaction between the stack memory and the DeltaNet-style state update is not presented as clearly as it could be. An explicit algorithm, step-by-step formulation, or architecture diagram would help readers better understand the computation. I think the token-mixer kernel can be simplely written as below equations(scalars are omitted). All my opinions are based on this formulation.

  $v_t^{read}, J_t = \text{StackMemory}(J_{t-1}, p_t)$

  $o_t, S_t = \text{DeltaRule}(S_{t-1}, q_t, k_t, v_t^{read} - v_t)$

- While the stack-memory perspective is interesting, the resulting token-mixer computation appears closely related to a dual-linear-attention structure. In particular, the formulation seems reminiscent of architectures such as GSA [1]. The paper would benefit from a more explicit discussion of the relationship between the proposed stack memory and GSA, including both conceptual and computational differences.

- Since DeltaStack introduces an additional memory computation per layer, its improvement over DeltaNet may partially stem from increased model capacity or computational budget rather than the stack mechanism itself. To better isolate the contribution of stack memory, it would be useful to include comparisons under approximately matched FLOPs, or at least a discussion of the computational tradeoffs.

[1] Gated Slot Attention for Efficient Linear-Time Sequence Modeling

---

> ### Author Rebuttal · Authors · 2026-03-29
>
> **Q:** The paper does not clearly explain how the output of the DeltaStack token mixer is computed.
>
> **A:** We agree that the presentation regarding the final output of the token mixer can be made more explicit.
> The formulation you pointed out is exactly correct:
>
> $J_{t}, v^{read}_{t}, = StackMemory(J_{t-1}, p_{t})$
>
> $S_t = DeltaRule(S_{t-1}, q_{t}, k_{t}, v_{t} - v^{read}_{t-1})$
>
> $o_t = S_t q_t$
>
> We currently formalize this mechanism through Equations 7 and 8. To prevent any ambiguity, we will incorporate your clear breakdown and add a dedicated architecture diagram in the revised version of the paper.
>
> **Q:** The paper would benefit from a more explicit discussion of the relationship between the proposed stack memory and GSA, including both conceptual and computational differences.
>
> **A:** We thank the reviewer for suggesting a comparison with the Gated Slot Attention (GSA) framework. Both models explore augmented memory, but they possess fundamental conceptual and structural differences:
> Conceptually, DeltaStack is mathematically derived from pushdown automata to achieve DCFL expressiveness. In contrast, GSA is designed to navigate the efficiency-retrieval tradeoff between the fixed-size state of linear attention and the unbounded KV cache of softmax attention.
> Architecturally, both methods can be viewed as "2-pass" linear attention updates—GSA operates as a 2-pass Gated Linear Attention (GLA), while DeltaStack is a 2-pass DeltaNet. However, the exact mechanics diverge significantly:
> * **Gating:** GSA utilizes the global forgetting gate from GLA; DeltaStack employs a local data-dependent decay via the Delta rule.
> * **Memory Role:** GSA uses an $\mathbb{R}^{m \times d}$ memory matrix (where $m$ is slot size) as the *final* readout memory. DeltaStack uses its $\mathbb{R}^{m \times d}$ stack ($J_t$) strictly as an *intermediate* structural memory that conditions the main associative memory ($S_t$).
> * **Data Structure:** GSA uses a content-based, unstructured slot memory updated via cumulative softmax. DeltaStack uses an explicit LIFO structured memory updated via differentiable push/pop scalar gates.
> * **Dimensionality:** GSA empirically requires a large slot size (e.g., $m=64$). DeltaStack operates effectively with a much smaller, lightweight stack (e.g., $m=8$ or $16$).
> * **Parameter Cost:** GSA introduces an additional learnable projection matrix that scales with $O(m)$. DeltaStack's parameter overhead is constant with respect to the stack size, requiring only a lightweight $O(d \times 3)$ projection for the action gates.
>
> We will incorporate this comparative analysis into the Related Work section.
>
> **Q:** To better isolate the contribution of stack memory, it would be useful to include comparisons under approximately matched FLOPs, or at least a discussion of the computational tradeoffs.
>
> **A:** We perform additional experiments to demonstrate the training throughput ([link](https://ibb.co/21dDjB6x)), memory cost ([link](https://ibb.co/tTZwTvyH)) and inference latency ([link](https://ibb.co/Nd9vcHNS)) of 1.3B parameter models on a single A100 GPU. DeltaStack maintains competitive scaling at long contexts, though it currently exhibits a moderate throughput drop and memory increase compared to DeltaNet. This overhead arises because our current implementation simulates stack push/pop operations by interleaving the input into an effective length of 2T and processing it with DeltaNet's dense kernels. Because half of this 2T sequence consists of zero-padded vectors during pop steps, developing custom, sparsity-aware kernels will bypass these redundant operations, offering a straightforward path to fully recover memory efficiency and peak throughput.

---

> > ### Author Rebuttal · Reviewer_1Rzt · 2026-04-03
> >
> > Thanks for your detailed response.
> > I acknowledge the effort you have made. But the overused computation and memory resources are still a concern.
> > Therefore, I decide to maintain my score.

---

### Official Review · Reviewer_xcSy · 2026-03-13

**Soundness:** 3
**Presentation:** 3
**Significance:** 2
**Originality:** 3
**Overall Recommendation:** 4
**Confidence:** 4

**Summary:**

This paper proposes DeltaStack, a novel recurrent architecture for language modeling. This model formulates traditional stack operations (push, pop, and read) as continuous, linear delta-rule updates to a specialized memory matrix. This formulation allows this stack-augmented network to be parallelized and, thus, make use of modern GPUs. Empirical results show superior performance on formal languages, and some improvements on ordinary language modeling.

**Compliance With Llm Reviewing Policy:**

Affirmed.

**Final Justification:**

This paper proposes an interesting and novel stack-based architecture based on the delta rule. Empirical results show that the proposed model can outperform vanilla DeltaNet (a popular RNN model based on the delta rule). During the rebuttal period, the authors have scaled up the scale of the experiments, strengthening the soundness of the paper. As a result, I have raised my score from 3 to 4.

**Key Questions For Authors:**

1. Can you provide empirical throughput and memory consumption of DeltaStack compared to standard DeltaNet and standard Transformer models? How much overhead does the $2T$ virtual step expansion and the Laplacian kernel computation actually add in practice?
2. You have claimed that the stack capacity can simply be increased at test time without retraining (Line 281). Have you validated this empirically?
3. The length extrapolation experiments show $S=8$ outperforming $S=16$ on CodeParrot. Does this suggest that forcing the model to "forget" or overwrite is actively beneficial for these benchmarks? Do you have any explanations/hypotheses for this result?

**Limitations:**

yes

**Strengths And Weaknesses:**

Strengths:

- The paper provides formal guarantees, explicitly demonstrating a constructive proof for Dyck$_{k,D}$ languages recognition in a single layer. This is a notable improvement over standard RNNs, SSMs, or Transformers which require deeper or wider circuits to handle nesting depth.
- DeltaStack nearly matches the idealized (but unscalable) Stack-RNN on hierarchical formal language tasks, validating the proposed mechanism. Also, the length extrapolation results on CodeParrot and OpenThoughts are strong.

Weaknesses:

- The language modeling experiments are constrained to 340M parameters trained on 15B tokens. While sufficient for a proof-of-concept, it remains unclear if the explicit LIFO stack inductive bias interacts favorably or detrimentally when scaling to multi-billion parameter LLMs trained on massive semantic datasets.
- Though the model is theoretically $O(LCd)$ in complexity, the paper lacks empirical throughput/latency measurements. Given the additional projections, softmax computations over the stack distance, and the $2T$ virtual steps mapping, empirical training/inference speed measurements compared to a standard DeltaNet and standard Transformer would provide a much clearer picture of overhead.

---

> ### Author Rebuttal · Authors · 2026-03-29
>
> **Q:** The language modeling experiments are constrained to 340M parameters trained on 15B tokens.
>
> **A:** We provide additional results on language modeling experiments scaling up to 760M parameter models, trained on 30B tokens from the FineWeb-Edu dataset.
> Compared to the DeltaNet baseline, DeltaStack (with a stack size $s=8$) consistently delivers improved perplexity and superior zero-shot performance across standard common-sense reasoning benchmarks.
> These results further validate the benefits and scalability of DeltaStack for larger-scale language modeling.
>
> | Model      | Wiki. ppl | LMB. ppl | LMB. acc | PIQA | Hella. | Wino. | ARC-e | ARC-c | Avg. |
> |------------|-----------|----------|----------|------|--------|-------|-------|-------|------|
> | DeltaNet   | 21.89     | 19.47    | 38.9     | 69.5 | 47.2   | 56.4  | 65.1  | 31.5  | 51.4 |
> | DeltaStack | 21.81     | 18.26    | 41.4     | 70.7 | 47.6   | 55.2  | 65.3  | 31.6  | 52.0 |
>
>
> **Q:** Can you provide empirical throughput and memory consumption of DeltaStack?
>
> **A:** We perform additional experiments to demonstrate the training throughput ([link](https://ibb.co/21dDjB6x)), memory cost ([link](https://ibb.co/tTZwTvyH)) and inference latency ([link](https://ibb.co/Nd9vcHNS)) of 1.3B parameter models on a single A100 GPU. DeltaStack maintains competitive scaling at long contexts, though it currently exhibits a moderate throughput drop and memory increase compared to DeltaNet. This overhead arises because our current implementation simulates stack push/pop operations by interleaving the input into an effective length of 2T and processing it with DeltaNet's dense kernels. Because half of this 2T sequence consists of zero-padded vectors during pop steps, developing custom, sparsity-aware kernels will bypass these redundant operations, offering a straightforward path to fully recover memory efficiency and peak throughput.
>
> **Q:** You have claimed that the stack capacity can simply be increased at test time without retraining (Line 281). Have you validated this empirically?
>
> **A:** Yes, we explicitly validate this claim empirically. In the formal languages experiments (Section 5.1, Table 1), and the accompanying analysis (Figures 2 and 3), we utilized this exact zero-shot capacity extension.
> As detailed in the Appendix (Lines 712-714), we trained the models with a stack size of $S=48$ and seamlessly extended it to $S=256$ during testing to match the evaluation protocol established by Deletang et al. (2023).
> DeltaStack's strong length generalization performance on DCF tasks confirms that stack capacity can be increased at test time without retraining or degrading accuracy.
>
> **Q:** The length extrapolation experiments show outperforming on CodeParrot. Does this suggest that forcing the model to "forget" or overwrite is actively beneficial for these benchmarks? Do you have any explanations/hypotheses for this result?
>
> **A:** We hypothesize that the length extrapolation performance gap shown in Figure 1, where a smaller stack ($S=8$) outperforms a larger one ($S=16$), is likely a regularization effect preventing overfitting.
> During pre-training on the FineWeb-Edu dataset, the model learns to map natural language to the stack. If we loosely model the hierarchical structure of a natural language document as a binary tree, a fully saturated stack of size $S=16$ could theoretically track a tree with $2^{16} \approx 65,000$ nodes.
> This depth is highly unrealistic for standard textual syntax, meaning the $S=16$ model may end up using the stack as a generic, unstructured buffer rather than a strict LIFO tracker. Consequently, this unstructured behavior fails to transfer robustly to the highly rigid, deeply nested syntax of code.
> Conversely, the constrained $S=8$ stack forces the model to aggressively "forget" stale nodes and prioritize immediate, local structural boundaries, which proves highly beneficial when extrapolating to long-context code.

---

> > ### Author Rebuttal · Reviewer_xcSy · 2026-04-01
> >
> > Most concerns are resolved, and I have raised by score accordingly.

---

### Official Review · Reviewer_MwUv · 2026-03-18

**Soundness:** 3
**Presentation:** 4
**Significance:** 3
**Originality:** 4
**Overall Recommendation:** 5
**Confidence:** 4

**Summary:**

DeltaStack augments DeltaNet (linear attention with delta rule) with a lightweight differentiable stack. The key insight: linear attention models have fixed-size state and thus cannot model recursive/hierarchical structures (DCFLs). DeltaStack adds a stack memory with a soft moving-pointer mechanism, where push/pop operations are formulated as rank-1 delta-rule updates — enabling full parallelization via DeltaNet's chunkwise algorithm. Theoretically proved to recognize Dyck-k languages (single layer). At 340M params/15B tokens, matches DeltaNet on standard LM benchmarks while showing better length extrapolation on code and math reasoning.

**Compliance With Llm Reviewing Policy:**

Affirmed.

**Key Questions For Authors:**

1. What happens when nesting depth exceeds S at test time? Does the model degrade gracefully or fail catastrophically?
2. What is the actual wall-clock overhead of the 2T virtual step expansion for the stack parallel algorithm?

**Limitations:**

1. LM gains are marginal. Table 3: DeltaStack (s=16) achieves 27.97 WikiText PPL vs DeltaNet 28.22 (Δ=0.25) and 46.2 avg zero-shot vs 45.5 (Δ=0.7). These differences are within noise for 340M models. The paper's main claim is expressivity, not LM performance, but the question remains: does hierarchical expressivity matter for practical LM?
2. Modular Arithmetic w/ Brackets remains unsolved: 35.2% (Table 1), barely above DeltaNet's 33.2%. The paper's own theory predicts this should be solvable (it's DCFL), but optimization fails. This gap between expressivity and learnability is acknowledged but not addressed.

**Strengths And Weaknesses:**

1. Elegant formulation. Casting push/pop as delta-rule updates to a memory matrix is genuinely clever. The unified update (Eq. 7) — Pop-Erase then Push-Erase as rank-1 perturbations — maps directly onto DeltaNet's chunkwise parallel algorithm. This means zero parallelism cost from the stack, unlike all prior stack-augmented models (Stack-RNN, Stack-Attention, etc.).
2. Formal language results are excellent. Table 1: 79.4% avg vs 65.5% for DeltaNet, approaching Stack-RNN's 96.3%. On DCF tasks specifically: Stack Manipulation 96.8%, Reverse String 98.8%, Dyck 94.3% — near-perfect hierarchical reasoning while maintaining parallelizability.

---

> ### Author Rebuttal · Authors · 2026-03-29
>
> **Q:** What happens when nesting depth exceeds S at test time?
>
> **A:** Similar to a standard bounded-size LIFO stack, the differentiable stack described in Eq. 7 would fail to perfectly track states if the required nesting depth exceeds the maximum capacity $S$ at test time.
> In such scenarios, the stack top $v^{read}_t$ retrieved by the soft pointer $p_t$ would become a superposition of multiple pushed vectors, which likely degrades the model's performance on "reasoning-dense" structural tasks.
> This aligns with prior theoretical work [1], which proves that a memory of size $O(m)$ is strictly necessary to reliably model a nesting depth of $m$.
> Our empirical analysis in Figure 2 (Section 5.4) further corroborates this: increasing the stack size $S$ consistently improves token accuracy on hierarchy-dependent tasks like reverse-string and stack-manipulation.
>
> **Q:** What is the actual wall-clock overhead of the 2T virtual step expansion for the stack parallel algorithm?
>
> **A:** We perform additional experiments to demonstrate the training throughput ([link](https://ibb.co/21dDjB6x)), memory cost ([link](https://ibb.co/tTZwTvyH)) and inference latency ([link](https://ibb.co/Nd9vcHNS)) of 1.3B parameter models on a single A100 GPU. DeltaStack maintains competitive scaling at long contexts, though it currently exhibits a moderate throughput drop and memory increase compared to DeltaNet. This overhead arises because our current implementation simulates stack push/pop operations by interleaving the input into an effective length of 2T and processing it with DeltaNet's dense kernels. Because half of this 2T sequence consists of zero-padded vectors during pop steps, developing custom, sparsity-aware kernels will bypass these redundant operations, offering a straightforward path to fully recover memory efficiency and peak throughput.
>
>
> **Q:** LM gains are marginal. Does hierarchical expressivity matter for practical LM?
>
> **A:** We agree that the current results on small-scale natural language modeling exhibit somewhat marginal gains.
> This is expected, as standard natural language tasks heavily emphasize semantic retrieval rather than deep structural recursion. In fact, even state-of-the-art LLMs often fail at seemingly trivial structural tasks like parity and matching closing parentheses [2].
> However, for "reasoning-heavy" or "structure-sensitive" domains such as code generation and mathematical reasoning, explicit hierarchical expressivity arguably plays an essential role.
> We provide preliminary evidence for this hypothesis via the robust length extrapolation results shown in Figure 1, where DeltaStack maintains a notably lower per-token loss than DeltaNet at sequence lengths far exceeding the training context.
>
> Furthermore, we provide additional results on language modeling experiments scaling up to 760M parameter models, trained on 30B tokens from the FineWeb-Edu dataset.
> Compared to the DeltaNet baseline, DeltaStack (with a stack size $s=8$) consistently delivers improved perplexity and superior zero-shot performance across standard common-sense reasoning benchmarks.
> These results further validate the benefits and scalability of DeltaStack for larger-scale language modeling.
>
> | Model      | Wiki. ppl | LMB. ppl | LMB. acc | PIQA | Hella. | Wino. | ARC-e | ARC-c | Avg. |
> |------------|-----------|----------|----------|------|--------|-------|-------|-------|------|
> | DeltaNet   | 21.89     | 19.47    | 38.9     | 69.5 | 47.2   | 56.4  | 65.1  | 31.5  | 51.4 |
> | DeltaStack | 21.81     | 18.26    | 41.4     | 70.7 | 47.6   | 55.2  | 65.3  | 31.6  | 52.0 |
>
> **Q:** Modular Arithmetic w Brackets remains unsolved.
>
> **A:** There are several reasons for DeltaStack's current limitations on the complex "Modular Arithmetic w/ Brackets" task.
> First, because DeltaStack couples its stack with the associative memory of DeltaNet, it inherits DeltaNet's baseline state-tracking profile. Table 1 shows that DeltaNet already performs modestly on the simpler "Modular Arithmetic w/o Brackets" task (66.4\% accuracy); thus, equipping it with a stack naturally limits its ceiling on the bracketed version. Second, while Theorem 4.4 proves a single-layer DeltaStack can recognize the Dyck language, it is unclear that a single layer can universally model the *entire* class of DCFLs. Third, even if expressivity is guaranteed, the optimization landscape for this specific task may be exceptionally difficult to traverse without specialized curricula or regularizers.
> As the primary contribution of this paper is introducing a lightweight, parallelizable, and expressive stack memory, resolving these finer-grained optimization challenges for complex arithmetic is left as an exciting direction for future work.
>
>
> [1] Hewitt, J., et al, "RNNs can generate bounded hierarchical languages with optimal memory"
>
> [2] Daking Rai, et al, "Failure by Interference: Language Models Make Balanced Parentheses Errors When Faulty Mechanisms Overshadow Sound Ones,"

---

> > ### Author Rebuttal · Reviewer_MwUv · 2026-04-01
> >
> > thanks for the detailed response.

---

### Decision · Program_Chairs · 2026-04-30

**Decision:**

Accept (regular)

**Comment:**

The paper introduces DeltaStack, which augments linear attention of DeltaNet with a differentiable stack memory. Reviewers rated soundness good-to-excellent, presentation good-to-excellent, significance fair-to-good, and originality good-to-excellent, reaching consensus on weak accept to accept (three weak accepts, one accept, one strong accept).

As main strength, reviewers mentioned that the paper proposes an elegant formulation of combining DeltaNet and differentiable stack memory. They specifically emphasized the significance of performance improvements on hierarchical formal language tasks and length extrapolation on code and math benchmarks.

All reviewers acknowledged the rebuttal as resolved or partially resolved, with several explicitly raising their scores after the new results.

All reviewers ranked paper with positive scores with recommendation to accept, and I concur.

The paper is technically sound with well-supported theoretical and empirical claims, strong formal-language results, and effective rebuttal addressing all major concerns, making its significance and originality sufficient to meet ICML acceptance criteria.